

# Estimating the national and regional prevalence of drinking or eating more than usual during childhood diarrhea in Malawi using the bivariate sample selection copula regression

Alfred Ngwira, Francisco Chamera and Matrina Mpeketula Soko

Basic Sciences Department, Lilongwe University of Agriculture and Natural Resources, Lilongwe, Malawi

## ABSTRACT

**Background:** Estimation of prevalence of feeding practices during diarrhea using conventional imputation methods may be biased as these methods apply to observed factors and in this study, feeding practice status was unobserved for those without diarrhea. The study aimed at re-estimating the prevalence of feeding practices using the bivariate sample selection model.

**Methods:** The study used 2015–2016 Malawi demographic health survey (MDHS) data which had 16,246 children records who had diarrhea or not. A bivariate Joe copula regression model with 90 degrees rotation was fitted to either drinking or eating more, with diarrhea as a sample selection outcome in the bivariate models. The prevalence of drinking more than usual and prevalence of eating more than usual were then estimated based on the fitted bivariate model. These prevalences were then compared to the prevalences estimated using the conventional imputation method.

**Results:** There was a substantial increase in the re-estimated national prevalence of drinking more fluids (40.0%, 95% CI [31.7–50.5]) or prevalence of eating more food (20.46%, 95% CI [9.87–38.55]) using the bivariate model as compared to the prevalences estimated by the conventional imputation method, that is, (28.9%, 95% CI [27.0–30.7]) and (13.1%, 95% CI [12.0–15.0]) respectively. The maps of the regional prevalences showed similar results where the prevalences estimated by the bivariate model were relatively higher than those estimated by the standard imputation method. The presence of diarrhea was somehow weakly negatively correlated with either drinking more fluids or eating more food.

**Conclusion:** The estimation of prevalence of drinking more fluids or eating more food during diarrhea should use bivariate modelling to model sample selection variable so as to minimize bias. The observed negative correlation between diarrhea presence and feeding practices implies that mothers should be encouraged to let their children drink more fluids or eat more food during diarrhea episode to avoid dehydration and malnutrition.

Corresponding author
Alfred Ngwira,
alfngwira@yahoo.com

## INTRODUCTION

Diarrhea in children is defined as the occurrence of three or more loose or liquid stools per day (*WHO, 2013*). Diarrhea is considered the second killer of under-five children after pneumonia causing an estimated 1.5 million under-five deaths every year (*UNICEF/WHO, 2009*). The prevalence of diarrhea in Malawi as of 2016 was 22% (*National Statistical Office of Malawi, 2017*). Diarrhea has a detrimental effect on child's nutrition status due to decreased food intake caused by anorexia (*Huffman et al., 1991*). To reduce deaths, dehydration and minimize the effects of diarrhea on nutritional status, mothers are encouraged to improve on feeding practices especially to give children more fluids and food than usual. The prevalence of feeding practices during childhood diarrhea in Malawi from the latest Malawi demographic health survey (MDHS) (*National Statistical Office of Malawi, 2017*) report is 31% for drinking more fluids and 13% for eating more food.

The estimation of prevalence of feeding practices during childhood diarrhea using demographic health survey (DHS) data is done by focusing on children who had diarrhea in the last two weeks. In this case, the prevalence of either drinking or eating more than usual is the number of children with diarrhea who drink or eat more than usual divided by the total number of children with diarrhea (*Croft, Marshall & Allen, 2018*). The use of complete case analysis (i.e., children who had diarrhea only) in the estimation of the feeding practices prevalence is biased since the selected sample is not random, that is, selection is more or less like voluntarily, since only children with diarrhea are considered. In addition, some participants who might have had responded that they did not have diarrhea, who are left out of the sample, might have said so due to fear of being ashamed of the disease particularly wealthier people who think they are always hygienic, while in actual sense they might have the disease and their feeding practice status would be better. This would result in underestimation bias of the prevalence. The use of children with diarrhea only can be considered as a missing data problem, where the feeding practice status is not observed for children without diarrhea. Standard or conventional approaches to dealing with missing data are single imputation, multiple imputation, inverse probability weighting or propensity score reweighting (*McGovern et al., 2015*). However if such methods can be applied in this study, they can be biased since they only account for observed factors and in this study, feeding practice status is not observed for those without diarrhea (*McGovern et al., 2015*). An alternative method which yield better estimates than the standard methods mentioned is the use of Heckman-style sample selection models (*Koné et al., 2019*). In this case, two equations are used, where one equation models the sample selection variable, for example, diarrhea status (yes/no) and another equation models the outcome variable of interest, for example, feeding practice status, for example, drinking more fluids or eating more food than usual.

Usually, the presence of diarrhea in a child is associated with poor household characteristics, for example, poor parental education according to *Asfaha et al. (2018)* (AOR = 2.88, 95% CI [1.70–4.88]) and *Manun'Ebo & Nkulu-wa-Ngoie (2020)* (AOR = 1.14, 95% CI [0.93–1.40]), and large household size according to *Omona et al. (2020)* (AOR = 7.185, 95% CI [1.353–38.147]). Such poor household characteristics in turn

are associated with poor feeding practices during diarrhea episode, for example, according to *Fikadu & Girma (2018)*, mothers who have one under-five child are 2 times (AOR = 2.11, 95% CI [1.38, 3.23]) more likely to have proper feeding practices during diarrhea episode as compared to those who have two and more under-five children. It is hypothesized therefore that there is a negative association between the presence of diarrhea and feeding practices (drinking or eating more) during childhood diarrhea.

The purpose of this study was to re-estimate the prevalence of drinking or eating more than usual during childhood diarrhea episode using sample selection copula bivariate model. The advantage of copula regression over the standard bivariate normal regression is that it offers flexibility in modelling dependence or association between two outcome variables (*Parsa & Klugman, 2011*). In this case, the copula regression can even model asymmetric dependence which is not the case with the standard bivariate normal distribution (*McGovern et al., 2015*). For example, in this study, since the hypothesized association between diarrhea presence and feeding practice status is negative, an exploration of asymmetric copula functions like the Joe copula with 90 degrees rotation which models the negative dependence was made. Specifically, the study sought to find out if diarrhea and drinking or eating more than usual were dependent and also the study aimed at re-estimating the prevalence of eating or drinking more than usual using the sample selection model. The significance of the study is that it would reveal if the estimates of prevalence of eating or drinking more during childhood diarrhea reported in the DHS report (*National Statistical Office of Malawi, 2017*) were biased or not due to the use of sample with diarrhea only. This would ensure correct policy making using the correct prevalences.

The following is how the article has been organized. First, methods regarding data and statistical analysis, are presented. This is followed by presentation of results, discussion and then conclusion.

## MATERIALS AND METHODS

### Data

The study used secondary data, the 2015–2016 Malawi DHS child record data. Permission to use the data was granted after asking for permission to use the data through the DHS web site (www.dhsprogram.com/data set_admin). The 2015–2016 MDHS study was ethically approved by Malawi Health Research Committee, Institutional Review Board of ICF Macro, Center for Disease and Control in Atlanta, GA, USA. No any other permission was needed to publish the results of the study. The MDHS was a two stage cluster sampling with stratification where clusters were stratified by residence (urban/rural) and then in each cluster, households were randomly selected (*National Statistical Office of Malawi, 2017*). In the first stage, 850 clusters, comprising of 173 clusters of urban areas and 677 clusters of rural areas were selected by probability proportional to size cluster sampling method. In the second stage, 30 households from each urban cluster and 33 households from each rural cluster were selected by systematic sampling. The data from households was then collected using the four questionnaires, that is, the woman, man, household and then biomarker questionnaire. This study used the child record data

mainly collected by the woman questionnaire. The response variables in the final data set were the presence of child diarrhea (yes/no) in the last two weeks and the feeding practice variables, namely, eating food more than usual (yes/no) and drinking fluids more than usual (yes/no). The independent variables were child age (in months), household size, maternal education (no education, primary, secondary, higher), water source (safe/unsafe), toilet (yes/no), house floor (cement/sand), district code of the child and interviewer identity code. The total number of records in the final data set was 16,246.

## Statistical analysis

A frequency distribution table of feeding practice in terms food and fluids intake during childhood diarrhea was made. A multiple variable bivariate sample selection model as defined by *Marra & Radice (2017a)* was then fitted, that is, if $y_{i1}$ is the sample selection outcome, that is, diarrhea presence (yes/no) and $y_{i2}$ is the second outcome observed only if child is selected, that is, has diarrhea, in this study for example, drinking more fluids or eating more food than usual. Then the flexible additive predictor of the bivariate model of diarrhea presence (yes/no) and drinking more or diarrhea presence (yes/no) and eating more food is defined as $\eta_{vi} = \beta_{v0} + \sum s_{vk,i}(z_{vk,i})$, $i = 1, 2, 3, \ldots, n$ where $v = 1, 2, c$. In addition, in the predictor, $\beta_0$ is the intercept and $s_{vk,i}(z_{vk,i})$ represents the generic effect of the independent variable and is specified according to the type of covariate considered. When $v = c$, the additive predictor $\eta_{ci}$ is for the copula parameter denoted by $\theta_i$, the dependence parameter between the two outcome variables involved. The bivariate distribution used in this study was the Joe copula with 90 degrees rotation which was opted for so as to model the negative dependence between diarrhea and the feeding practice variables as per the hypothesis, and after the standard bivariate normal copula revealed a negative dependence between the two outcome variables. The range of the copula parameter with this bivariate distribution was $(-\infty, -1)$ where $\theta = -1$ means independence and dependence otherwise. The main objective in this study was to estimate the prevalence of the observed outcome, $y_{i2}$, given the selected sample, for example, the prevalence of drinking more fluids or eating more food during diarrhea episode defined as $P(Y_2 = 1)$. This was computed by the formula: $P(Y_2 = 1) = \frac{\sum_{i=1}^{n} w_i[1 - F_2(\hat{\eta}_{i2})]}{\sum_{i=1}^{n} w_i}$, where $w_i$ were the survey weights, the woman individual sample weights. Model estimation was by penalized maximum likelihood estimation (PMLE) considering that the usual maximum likelihood estimation (MLE) could lead to over fitting due to the presence of smooth functions (*Filippou, Marra & Radice, 2017*). Model fitting, prevalence estimation and mapping of regional prevalence was implemented by the GJRM package in R (*Marra & Radice, 2017b*).

## RESULTS

Table 1 presents the percentage of children under-five in terms of feeding practices during diarrhea as reported from MDHS report (*National Statistical Office of Malawi, 2017*). Thirty one percent of children with diarrhea were given more than usual fluids and thirty four percent were given less fluids which is a concern. Five percent of the sick

**Table 1 Percentage of feeding practices during diarrhea.**

|  | Feeding practice | | | | |
|---|---|---|---|---|---|
|  | More | Usual/same | Less | None | Never gave |
| Liquids | 31 | 30 | 34 | 5 | 0 |
| Food | 13 | 32 | 43 | 6 | 5 |

children were not given any fluids. Sixty one percent of the sick children were given recommended liquid as compared to fourty five percent children that were given recommended food. These percentages were used as a bench mark as the new percentages using the sample selection model were estimated. The focus however was on the percentage of children feeding or drinking more than usual during diarrhea.

After fitting the bivariate sample selection model using the Joe copula, the estimated national prevalence of drinking more fluids during childhood diarrhea using the conventional imputation method was 28.9% with 95% confidence interval as (27.0, 30.7). Adjusting for sample selection, using the sample selection bivariate model, the estimated prevalence of drinking more fluids was 40.0% with 95% confidence interval (31.7, 50.5). Regarding eating more food, the estimated national prevalence of eating more during diarrhea using the conventional imputation method was 13.1% with 95% confidence interval (12.0, 15.0), and the estimated prevalence using the bivariate sample selection model was 20.46% with 95% confidence interval (9.87, 38.55). The estimated average copula parameter for the bivariate model of diarrhea and drinking more fluids was −1.23 with 95% confidence interval as (−1.53, −1.09) and that of diarrhea and eating more food was −1.29 with 95 % confidence interval as (−2.57, −1.04), both showing negative dependence. The association of drinking and eating more with diarrhea may be considered as weak since the copula parameter values are close to −1.

Figure 1 presents the map of prevalence of drinking more fluids during childhood diarrhea by region. The prevalence estimates by the bivariate sample selection model (Fig. 1B) are relatively higher than the imputation based estimates (Fig. 1A). The copula parameter shows relatively strong negative dependence between diarrhea and drinking more fluids in many districts (Fig. 1C). Figure 2 shows the distribution of prevalence of eating more food. The prevalence estimates by the bivariate sample selection model (Fig. 2B) are also relatively higher than the imputation based estimates (Fig. 2A). The copula parameter distribution shows many areas having weak dependence between diarrhea and eating more food (Fig. 2C).

## DISCUSSION

The study has looked at the re-estimation of the prevalence of feeding practices during childhood diarrhea by focusing on drinking and eating more than usual. The standard estimation procedure is prone to be biased as the final selected sample is not random as individuals are selected depending on whether they have diarrhea or not. The study therefore tried to investigate the degree of biasness attached to the estimates if the standard method is used, by re-estimating the prevalences using the novel approach which takes

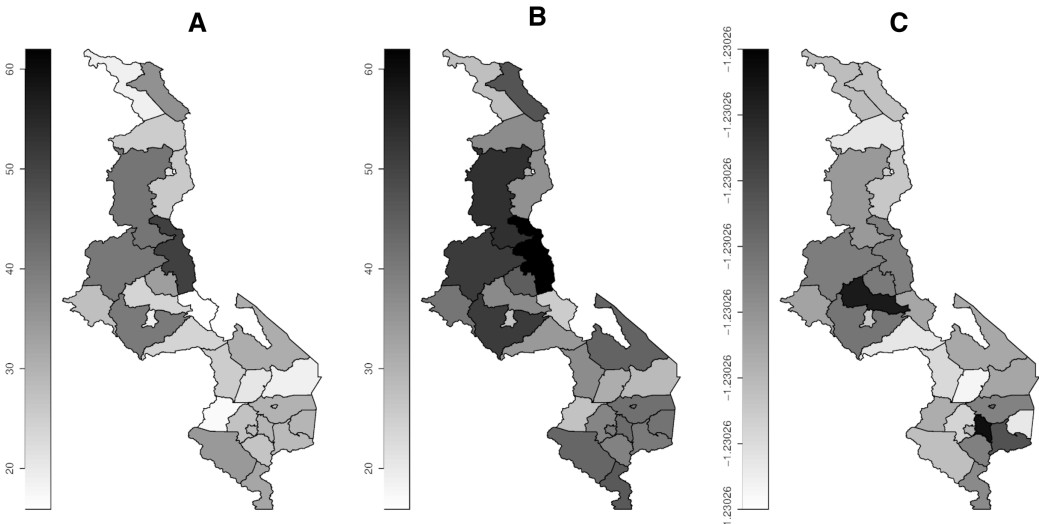

**Figure 1 Prevalence map of drinking more fluids by district.** (A) Imputation model. (B) Sample selection model. (C) Copula parameter. Darker (high), gray (middle) and white (low).

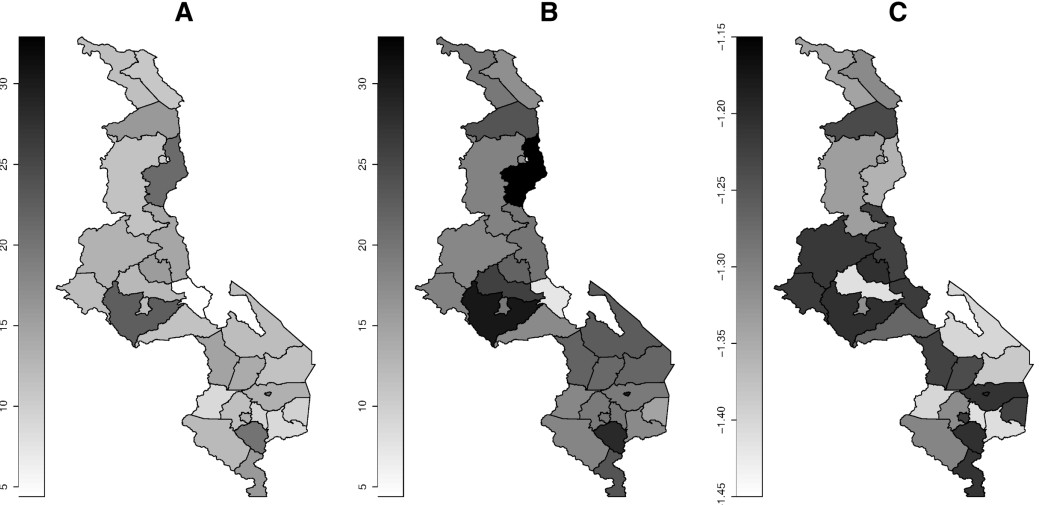

**Figure 2 Prevalence map of eating more food.** (A) Imputation model. (B) Sample selection model. (C) Copula parameter. Darker (high), gray (middle) and white (low).

into account sample selection process. Sample selection process was taken into account in estimation by using the bivariate sample selection model where one of the two equations, modeled sample selection variable, diarrhea status (yes/no).

The study found that the prevalence estimates of both drinking more liquids and eating more food using the bivariate model were substantially greater than those estimated by the conventional, imputation method. The increased prevalence was likely, since one international study (*Bani et al., 2002*) found much higher prevalence of mothers

increasing the volume of fluids given during childhood diarrhea episode (75.5%). The observed differences in the prevalence estimated by the bivariate sample selection model and the usual imputation method might be due to the non-randomness of the sample selected (*Marra & Radice, 2017a*). Non-randomness of the sample would be as a result of selection process, as samples were being selected if they had diarrhea, otherwise they were not selected. The estimates by bivariate sample selection model would be with minor bias as non-randomness was corrected by modelling sample selection variable (diarrhea status) in addition to modelling the outcomes of interest, feeding status variables. The increase in prevalence estimate from standard estimation to sample selection estimation may be due to the fact that those not selected (without diarrhea) were likely to have good education and small number of children since low education and large household size are positively correlated with diarrhea (*Asfaha et al., 2018*) and this would mean increased drinking and eating more in case sample selection process is corrected as increased education and small household size increase the intake of fluids and food (*Fikadu & Girma, 2018*). An indication of weak correlation between drinking and eating more food and diarrhea is consistent with *Huffman et al. (1991)* where having diarrhea did not significantly affect eating habits though there was a decline in eating food as children appetite reduced.

The distribution of estimated prevalence of drinking or eating more by region (Figs. 1 and 2B) shows that there is less variation and most regions especially around the cities show increased intake of fluids and food. Generally, there is reduced intake of food compared to fluids. The reduction in food intake may be due to decrease in appetite in the diarrhea patients as explained by *Huffman et al. (1991)* and *Paintal & Aguayo (2016).* Similar distribution of fluid and food intake across all the regions may due to the fact that all regions might have similar spatial determinants of feeding practices during diarrhea, for example, average household size is seen to be similar across the three main regions of Malawi with the following average household sizes: 3.7 (north), 3.6 (central) and 3.7 (south) (*National Statistical Office of Malawi, 2017*), and household size has been found to be associated with feeding practices (*Fikadu & Girma, 2018*).

## CONCLUSION

The study finds that both estimated prevalence of drinking and eating more food by the standard method were likely to be biased considering that they deviated greatly from the estimate based on the sample selection model found in this study. There is weak negative correlation between the presence of diarrhea and the feeding practices during diarrhea. The implication of the results is that the prevalence of drinking or eating more food during childhood diarrhea should be estimated by taking into account sample selection process (diarrhea presence) so as to correct for the biasness that may arise due to non-randomness of the sample.

## ACKNOWLEDGEMENTS

We thank the demographic health survey (DHS) for providing the data that was used.

### Funding

The authors received no funding for this work.

### Competing Interests

The authors declare that they have no competing interests.

### Author Contributions

- Alfred Ngwira conceived and designed the experiments, performed the experiments, analyzed the data, prepared figures and/or tables, and approved the final draft.
- Francisco Chamera performed the experiments, analyzed the data, authored or reviewed drafts of the paper, and approved the final draft.
- Matrina Mpeketula Soko conceived and designed the experiments, performed the experiments, authored or reviewed drafts of the paper, and approved the final draft.

### Data Availability

Data and code are available in the Supplemental Files.

### Supplemental Information

Supplemental information for this article can be found online at http://dx.doi.org/10.7717/peerj.10917#supplemental-information.

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
