# Peer review of "Estimating the national and regional prevalence of drinking or eating more than usual during childhood diarrhea in Malawi using the bivariate sample selection copula regression"

_PeerJ, doi:10.7717/peerj.10917_

## Round 0.1 · original submission · Major Revisions

Please carefully consider all the comments from the reviewers. In particular, I strongly suggest that you use a native English speaker for reviewing the test since currently, some sentences are hard to understand.

Reviewer 1 ·

Basic reporting

this is a very good applied work that uses an advanced method to solve an important applied problem. I do not see any major issue here apart from the fact that the authors need to reference properly the methods used, specifically they have to cite Marra et al JASA 2017 and the GJRM package which the authors used for the analysis

Experimental design

It looks fine as far as I can see

Validity of the findings

They look robust as far as I can see

Additional comments

very good work, please fix the references

·

Basic reporting

27-28 It is known that conventional methods may be biased. Please elaborate more about the conventional methods and why it is biased. By simply reading the background part, the readers may have more questions about how the samples are selected in conventional methods in detail and the impact of the results.

34 Please write this sentence clearer and more concise.

39 Please comment more on the maps of prevalences

44 The weakly association of diarrhea and drinking or eating is an impressive finding

47 Readers may be interested in knowing more about what kind of strategies

59-63 I appreciate your investigation of diarrhea data, however, the structure of these sentences is monotonous. Please try to use another way to rephrase these sentences.

67 2015 to 2016 would be a better wording compared to 2015/2016

70 Readers would like to see more about how DHS data was selected. Please use more citations instead of assuming “ that is, selection into the sample is based on whether the child has diarrhea or not”

72-78 I appreciate your citations for these examples, however, I suggest that confidence interval or P-value be included from these studies to show a statistically significant result. For eg, Asfaha et al 2018, saying households with ≥3 number of children under-five [AOR = 4.05, 95% CI (1.91, 8.60)] was associated with childhood diarrhea

76-78 A good hypothesis

79-81 A bit ambiguous of the logic behind it. Please rephrase the sentence to make it clearer. Do you mean prevalence would be underestimated due to diarrhea status when children were selected for the study?

85 Why does the flexibility in marginal distribution benefit your study? Please comment more on this part.

152 I suggest using the same ‘diarrhea’ consistently throughout the article

155 Please elaborate on the alpha level of the confidence interval that you use

159 13.1 is a percentage

193 Please explain more about how nonrandomness was corrected.

Data and tables look good. Here are some minor issues. A lot of sentences are written in the passive voice. I suggest that rephrase these sentences and make them in an active voice. For eg. Change 'something was collected' to 'we collected something'. Besides, there are some minor spelling issues. Sentences should be written more scientifically, avoiding using words 'say XXXX', 'let xxx be', etc.

Experimental design

The research goal meets the aims and scope of the journal. The hypothesis is creative and the results are impressive. However, there is still space for improvement. For example, in the beginning, the article should convey more explicitly how the conventional method was biased in detail, and make it clearer how this flaw led to your hypothesis.

Validity of the findings

214 Please follow the order, conclusion, and then discussion

In the discussion part, I would suggest adding more insights on future changes/policy-making/advocates/solutions as mentioned in the beginning.

Additional comments

Overall, the article is well written. Hope to see Scientific English Language improvement in the future.

·

Basic reporting

1) The author needs to cite more recent published papers. Currently, there's no cited paper published after 2018.

2) The English language is a big problem. The author should make a careful reversion to ensure that an international audience can clearly understand the text.
For example:
L28, "due to selection procedure." -> "due to the selection procedure."
L30, "non randomness of sample" -> "non-randomness of the sample"
L60, "Diarrhea is considered as the second" -> "Diarrhea is considered the second"
L60, “under five” -> “under-five”
L64, “minimise” -> “minimize”
L72, "house hold" -> "household"
L75, "with large number" -> "with a large number"
L77, "there is negative association" -> "there is a negative association"
L166, "diarrhoea" ->"diarrhea"

Similar problems exist in the remaining paper. The author needs to fix all of them.

Experimental design

The author needs to explain the formula in L125. The meaning of those symbols is not very clear.

Validity of the findings

1) Fig. 1 (c) keeps the first five digits after the decimal point. At the meanwhile, Fig. 2 (c) holds the first two digits after the decimal point. The numbers in Fig. 1 (c) are very close to each other, which contrasts the claim in L166 "... copula parameter for the bivariate model of diarrhoea and drinking more fluids was -1.23 with confidence interval (-1.53,-1.09)."

2) In the caption, "Dark (high) and white (low)." However, Fig. 1 (c) and Fig. 2 (c) do not follow it.

3) The color of the figure is not illustrative. The author needs to make the color more contrast when the range of the number is wider.

4) It would be better if the author could use numbers to evaluate the prevalence in different regions in addition to figures.

---

## Round 0.2 · accepted · Accept

I think the manuscript has improved with the modifications included by the authors and it is now acceptable for publication.

·

Basic reporting

I would suggest using "diarrhea" instead of "diarrhoea" in table 1 to keep it consistent.

Experimental design

well written

Validity of the findings

well written

Additional comments

English writing has improved since last time. The whole article looks fluent and consistent and easy to read. However, some minor issues need to be fixed.